# Changes in Cellular Localization of Inter-Alpha Inhibitor Proteins after Cerebral Ischemia in the Near-Term Ovine Fetus

**DOI:** 10.3390/ijms221910751

**Published:** 2021-10-04

**Authors:** Kazuki Hatayama, Boram Kim, Xiaodi Chen, Yow-Pin Lim, Joanne O. Davidson, Laura Bennet, Alistair J. Gunn, Barbara S. Stonestreet

**Affiliations:** 1Department of Pediatrics, Women & Infants Hospital of Rhode Island, Alpert Medical School, Brown University, Providence, RI 02905, USA; KHatayama@wihri.org (K.H.); Boram.Kim@Pennmedicine.upenn.edu (B.K.); xchen@wihri.org (X.C.); 2ProThera Biologics, Inc., Providence, RI 02903, USA; yow-pin_lim_MD@brown.edu; 3Department of Pathology and Laboratory Medicine, Alpert Medical School of Brown University, Providence, RI 02903, USA; 4Department of Physiology, The University of Auckland, Auckland 1142, New Zealand; joanne.davidson@aukland.ac.nz (J.O.D.); l.bennet@auckland.ac.nz (L.B.); aj.gunn@auckland.ac.nz (A.J.G.)

**Keywords:** brain injury, hypoxia-ischemia, inter-alpha inhibitor proteins, ovine fetus

## Abstract

Inter-alpha Inhibitor Proteins (IAIPs) are key immunomodulatory molecules. Endogenous IAIPs are present in human, rodent, and sheep brains, and are variably localized to the cytoplasm and nuclei at multiple developmental stages. We have previously reported that ischemia-reperfusion (I/R) reduces IAIP concentrations in the fetal sheep brain. In this study, we examined the effect of I/R on total, cytoplasmic, and nuclear expression of IAIPs in neurons (NeuN^+^), microglia (Iba1^+^), oligodendrocytes (Olig2^+^) and proliferating cells (Ki67^+^), and their co-localization with histones and the endoplasmic reticulum in fetal brain cells. At 128 days of gestation, fetal sheep were exposed to Sham (*n* = 6) or I/R induced by cerebral ischemia for 30 min with reperfusion for 7 days (*n* = 5). Although I/R did not change the total number of IAIP^+^ cells in the cerebral cortex or white matter, cells with IAIP^+^ cytoplasm decreased, whereas cells with IAIP^+^ nuclei increased in the cortex. I/R reduced total neuronal number but did not change the IAIP^+^ neuronal number. The proportion of cytoplasmic IAIP^+^ neurons was reduced, but there was no change in the number of nuclear IAIP^+^ neurons. I/R increased the number of microglia and decreased the total numbers of IAIP^+^ microglia and nuclear IAIP^+^ microglia, but not the number of cytoplasmic IAIP^+^ microglia. I/R was associated with reduced numbers of oligodendrocytes and increased proliferating cells, without changes in the subcellular IAIP localization. IAIPs co-localized with the endoplasmic reticulum and histones. In conclusion, I/R alters the subcellular localization of IAIPs in cortical neurons and microglia but not in oligodendrocytes or proliferating cells. Taken together with the known neuroprotective effects of exogenous IAIPs, we speculate that endogenous IAIPs may play a role during recovery from I/R.

## 1. Introduction

Inter-alpha Inhibitor Proteins (IAIPs) are immunomodulatory proteins that are part of the innate immune system. IAIPs inhibit destructive serine proteases, reduce proinflammatory cytokines, increase anti-inflammatory cytokine production, attenuate complement activation during inflammation, and neutralize the cytotoxicity of extracellular histones [1,2,3,4,5,6]. They are synthesized mainly in the liver and are found in high levels in the plasma of adults and newborns, which suggests that they are essential proteins [1,5,7]. The two major forms found in human plasma are Inter-alpha Inhibitor (*Ia**I*, molecular weight (MW) = 250 kDa), consisting of two heavy chains (H1 and H2) and a single light chain termed bikunin (or urinary trypsin inhibitor, MW = 30 kDa), and Pre-alpha Inhibitor (*PaI*, MW =125 kDa), consisting of one heavy (H3) and one light chain (bikunin) [1]. After systemic inflammation, the heavy chain H2 and light chain are down-regulated and the relevant molecules (*IaI*) act as negative acute-phase proteins, whereas the H3 chain is up-regulated and the corresponding *PaI* molecule is a positive acute-phase protein [8].

Endogenous IAIPs have been found in most peripheral tissues, including lung, pancreas, liver, kidney, intestine, and connective tissues, and in the brain [9,10,11,12,13,14,15]. IAIPs are present in brain tissue, choroid plexus, and cerebral spinal fluid (CSF) over a wide span of ovine development as both the 125 kDa *PaI* and 250 kDa *IaI* protein moieties [16]. Further, we have identified endogenous IAIP genes and proteins in cultured mouse neurons, proteins in cultured rat neurons, microglia, and astrocytes, and in vivo in neurons, microglia cells, and astrocytes, as well as in multiple brain regions [17]. More recently, ubiquitous immunoreactivity of IAIPs has been reported in the human cerebral cortex from early in development through the neonatal period and in adults, in neurons and astrocytes [18]. The immunoreactivity of IAIPs was predominately localized in the nucleus at all ages, but cytoplasmic IAIP expression was more abundant in adulthood than in younger ages [18]. Although IAIPs are ubiquitous in the brains of sheep, rodents, and humans, information regarding the function of these proteins in normal brains is not as yet available.

Hypoxic-ischemic (HI) brain injury is the one of the most common and severe neurological problems in the perinatal period [19]. HI-related brain injury occurs in approximately one to three per 1000 full-term and near-term births, resulting in moderate to severe encephalopathy [20,21]. This is associated with a high incidence of mortality, morbidity, and disabilities in survivors [19]. After HI-related injury to the brain, there is a progressive evolution of the brain damage over hours to days after the insult [22,23,24]. Inflammation is increasingly identified as an important contributor to the evolution of injury in the developing brain [22,25,26].

In term-equivalent fetal sheep, ischemia with reperfusion (I/R) is associated with acute decreases in the expression of *IaI* and *PaI* moieties in the cerebral cortex and in the cerebellum of fetal sheep four hours after ischemia, their expression levels return toward control values at 24 and 48 h after ischemia [27]. Although there is a paucity of information regarding the attributes of endogenous IAIPs in the brain, recent evidence demonstrates that the Inter-alpha-trypsin inhibitor heavy chain 4 is decreased in the serum of human patients after acute ischemic stroke [28].

There is growing evidence that IAIPs play an important role in inflammation and that inflammation represents a critical component of neonatal HI brain damage [1,3,4,22]. We therefore postulated that endogenous IAIPs in the brain may be affected by HI-related inflammation in the fetal and neonatal brain. We have previously shown changes in the subcellular localizations of IAIPs after HI [29]. However, in our previous study, we did not examine whether there were differences between types of cells after HI [29].

Therefore, the objectives of the current study were to examine the immunohistochemical expression of IAIP localization in neurons, microglia, and oligodendrocytes in the brains of near-term fetal sheep after exposure to I/R followed by seven days of recovery. We examined the presence of IAIPs in proliferating cells because we have previously shown that the Ki67 proliferative antigen marker was increased in sprouting-type microvessels and in astrocytes after cerebral ischemia in near-term fetal sheep and in periventricular white matter after exposure to asphyxia in preterm fetal sheep a [30,31] and after cerebral ischemia in near-term fetal sheep [32]. In addition, we examined the co-localization of IAIPs in the nucleus with histones and the endoplasmic reticulum with calnexin in order to better define the localization of IAIP expression within the subcellular compartments after ischemia. Elucidation of fundamental changes in HI-related brain injury could help to uncover key mechanisms of injury that could be targeted to develop novel therapeutic treatments.

The brain maturation of the sheep fetus at a gestational age of 0.85 is broadly similar to the full-term human infant, and so is relevant to brain injury in the human newborn [33,34]. The experimental paradigm involves bilateral carotid artery occlusion followed by prolonged reperfusion for seven days, and results in a consistent watershed pattern of brain injury [35]. This is consistent with one of the major clinical patterns observed in infants after exposure to HI brain injury [36]. The evolution of injury is well described and consistent with clinical timing, as shown by the successful translocation of therapeutic hypothermia from this paradigm to clinical care [37].

## 2. Results

### 2.1. Validation of R-22c pAb Specificity by Western Immunoblot

Western immunoblot validation of the specificity of the polyclonal rabbit anti-rat IAIPs antibody (R-22c pAb, ProThera Biologics) is shown in Figure 1. The R-22c pAb detected positive bands for the 250 (*IaI*) and 125 kDa (*PaI*) IAIP in the purified human IAIP (lane A) and detected two specific bands in the sheep serum (lane B). Moreover, the R-22c pAb detected the two bands in tissue extract from the sheep brain (lane C). By contrast, the pre-immune serum used as a negative control for the R-22c pAb did not recognize IAIP bands. This shows that the R-22c pAb has good specificity for IAIPs in sheep serum and brain tissue.

### 2.2. Subcellular Localization of IAIP Shifts from the Cytoplasm to the Nucleus in the Cerebral Cortex but Not in White Matter of Fetal Sheep after I/R Injury

Immunofluorescence staining with the specific anti-IAIP antibody (R-22C pAb) identified positively stained cells for IAIPs in randomly selected areas from representative cerebral cortical and white matter regions in the samples from fetal sheep brains in Sham controls and after exposure to I/R (Figure 2A). Consistent with our previous findings in the human brain during development [18], more than 90 percent of cells exhibited positive immunofluorescent staining for IAIPs in both the cerebral cortex and white matter of the Sham and I/R exposed fetal sheep (Figure 2A,B). The positive immunofluorescent staining for IAIPs appeared more prominent in the cytoplasmic compartment of the cerebral cortex (IAIPs and Merged—indicated by arrows) of the Sham, and in the nuclear compartment of the I/R group (IAIPs and Merged—indicated by arrows) because the IAIP immunoreactivity was co-localized with the DAPI-stained nuclei in the cerebral cortex of I/R-exposed fetal sheep (Figure 2A). Immunofluorescent staining for IAIPs appeared to be mainly localized to the nuclear compartment in the white matter of both the Sham and I/R-exposed (IAIPs and Merged—indicated by arrows) fetal sheep (Figure 2A).

The number of IAIP-positive cells in the cytosolic and nuclear compartments were determined as a percentage of the total number of cells. The total number of cells was determined by counting all DAPI-positive nuclei within the cerebral cortex and white matter of the Sham and I/R exposed fetal sheep (Figure 2B–D). The fluorescent expression of the total IAIPs was greater than 90 percent of all cells in the cerebral cortex and white matter of the Sham and I/R exposed fetal sheep (Figure 2B; repeated measures ANOVA, F (1,9) = 0.02, *n* = 22, *p*= 0.89) and did not differ between the groups in the cerebral cortex (Figure 2B; Tukey honestly significant difference (HSD), *p* = 0.87) or white matter (Figure 2B; Tukey HSD: *p* = 0.7692). In contrast, there were differences in the cytoplasmic localization of IAIPs between the cerebral cortex and white matter in each group (Figure 2C; repeated measures ANOVA, F (1,9) = 242.87, *n* = 22, *p* = 0.0000). The IAIP-positive fluorescent stained cells in the cytoplasm were significantly greater in number (Figure 2C; Tukey HSD, *p* = 0.0002) in the cerebral cortex of the Sham group compared with the I/R-exposed fetal sheep but were present in less than 5% of the total number of cells (Sham: 1.46%, I/R: 1.71%) in the white matter. 

Moreover, there were differences in the nuclear localization of IAIPs between the cerebral cortex and white matter in each group (Figure 2D; repeated measures ANOVA, F (1,9) = 122.88, *n* = 22, *p* = 0.0000). The percentage of IAIP-positive cells in the nuclear compartment was greater (Figure 2D; Tukey HSD, *p* = 0.0002) in the cerebral cortex of the I/R than in the Sham-exposed fetal sheep, but did not differ (Figure 2D, Tukey HSD, *p* = 0.988) in the white matter between groups. Taken together, these findings suggest that the localization of IAIPs changed from cytoplasmic predominance in the Sham to nuclear predominance after exposure to I/R in the cerebral cortex of the fetal sheep. By contrast, subcellular localization did not change in the white matter after exposure to I/R.

### 2.3. Expression of Neurons, and Subcellular IAIP Localization in Neurons in the Cerebral Cortex of the Sham- and I/R-Exposed Fetal Sheep

Neuronal staining was analyzed by immunofluorescent Fox3/neuronal nuclei (NeuN) detection with mouse anti-monoclonal antibody in the fetal sheep brains (Figure 3A). Similar to the findings for the single stained IAIP-positive cells (Figure 2), the NeuN-positive cells also co-stained positive for IAIPs (Figure 3A; IAIPs and Merged—indicated by arrows). As expected, the total number of NeuN-positive neurons (expressed as a percentage of the total number of DAPI-positive cells) was reduced (Figure 3B; Two Group T-Test, *p* = 0.00005) after exposure to I/R [35]. The number of the total IAIP-positive neurons was normalized to the number of NeuN-positive neurons (Figure 3C–E). There were differences in the localization of IAIPs between total, cytoplasmic, and nuclear compartments of the neurons between the two groups. The total number of IAIP-positive NeuN-positive neurons was more than 80 percent in the cerebral cortex of both the Sham and I/R groups and did not differ between the groups (Figure 3C; Two Group T-Test, *p* = 0.1476). However, there were more cytoplasmic IAIP-positive NeuN-positive cells in the cerebral cortex of the Sham group than in the I/R group (Figure 3D; Two Group T-Test, *p* = 0.0091), but the number of nuclear IAIP-positive NeuN-positive cells were not significantly different between the groups (Figure 3E; Two Group T-Test, *p* = 0.0566). Taken together, these findings suggest that the number of cytoplasmic IAIP-positive NeuN-positive cells decreased after exposure to I/R.

### 2.4. Expression of Iba1 Stained Microglia, and Subcellular IAIP Localization in Microglia in the Cerebral Cortex of the Sham- and I/R-Exposed Fetal Sheep

Iba1-stained cells increased as a percentage of DAPI-positive cells, as expected, after exposure to I/R (Figure 4B; Two Group T-Test, *p* = 0.0000) [38]. The Iba1-positive microglia also co-stained for IAIPs (Figure 4A; IAIPs and Merged—indicated by arrows). The IAIP-positive Iba1-positive cells were normalized to the total number of Iba1-positive cells (Figure 4C–E). There were differences in the localization of IAIPs between total, cytoplasmic, and nuclear compartments in microglia between the two groups (Figure 4). There were more IAIP-positive Iba1 cells in the cerebral cortex of the Sham than in the I/R fetal sheep (Figure 4C; Two Group T-Test, *p* = 0.0002). The numbers of cytoplasmic IAIP-positive Iba1-positive stained cells were negligible in the cerebral cortex of both groups of fetal sheep and did not differ between the groups (Figure 4D, Two Group T-Test, *p* = 1.0000). In contrast, similar to the staining for the total Iba1-positive IAIP-positive cells, the nuclear IAIP-positive Iba1-positive staining was greater in the cerebral cortex of the Sham group compared with the I/R group (Figure 4E; Two Group T-Test, *p* = 0.0002).

### 2.5. Expression of Olig2-Stained Oligodendrocytes and Subcellular IAIP Localization in Oligodendrocytes in the White Matter of the Sham- and I/R-Exposed Fetal Sheep

Olig2-positive oligodendrocytes, shown as a percentage of DAPI-positive cells, decreased after exposure to I/R (Figure 5B; Two Group T-Test, *p* = 0.0000) [39]. The Olig2-positive oligodendrocytes also co-stained for IAIPs (Figure 5A; IAIPs and Merged—indicated by arrows). The IAIP-positive Olig2-positive oligodendrocytes were normalized to the total number of Olig2-positive oligodendrocytes (Figure 5C–E). There were no differences in the localizations of IAIPs between the total, cytoplasmic, and nuclear compartments in oligodendrocytes between the groups (Figure 5C–E). Almost all of the Olig2-positive cells also co-stained for IAIPs in the white matter of the Sham- and I/R-exposed fetal sheep (Figure 5C; Two Group T-Test, *p* = 0.6204). On the other hand, there was negligible IAIP staining in the cytoplasm of the Olig2-positive oligodendrocytes in the white matter of both groups (Figure 5D; Two Group T-Test, *p* = 0.4402). Most of the IAIP-positive staining was confined to the nuclei of the Oligo2-positive oligodendrocytes in both the Sham- and I/R-exposed fetal sheep (Figure 5E; Two Group T-Test, *p* = 0.9396).

### 2.6. Subcellular Localization of IAIPs in Ki67 Positive Proliferating Cells in the Cerebral Cortex of the Sham and I/R-Exposed Fetal Sheep

Consistent with our previous reports, the number of proliferating Ki67-positive cells normalized to the total number of DAPI-positive cells was higher in the cerebral cortex of the I/R- than the Sham-exposed fetal sheep (Figure 6B; Two Group T-Test, *p* = 0.0009) [30,31]. The proliferating Ki67-positive stained cells also co-stained for IAIPs (Figure 6A, IAIPs and Merged—indicated by arrows). There was no difference in the localization of IAIPs between total (Figure 6C; Two Group T-Test, *p* = 0.3969), cytoplasmic (Figure 6D; Two Group T-Test, *p* = 0.0742), and nucleus (Figure 6E; Two Group T-Test, *p* = 0.1989) compartments in proliferating cells between the two groups. The Ki67-positive stained cells uniformly stained for IAIPs in the cerebral cortex of both groups of fetal sheep (Figure 6C). The IAIP-positive stained Ki67 cells stained mainly in the nuclei rather than in the cytoplasm of the proliferating cells in the cerebral cortex of the Sham and HI groups (Figure 6D,E).

### 2.7. Co-Localization of IAIPs with Histones in Nucleus and Calnexin in the Endoplasmic Reticulum of the Cerebral Cortex in the Sham- and I/R-Exposed Fetal Sheep

Although we have previously demonstrated immunohistochemical expression of IAIPs both in the nucleus and cytoplasm in the cerebral cortex of post-mortem human brain tissue obtained from preterm and full-term neonates, and adult subjects [18], we sought to further verify the detailed localization of the immunohistochemical expression of IAIPs with the classical nuclear marker, histones (Figure 7) and calnexin, a marker of the endoplasmic reticulum (Figure 8).

Histone positively-stained cells were detected in the Sham- and I/R-exposed fetal sheep, as expected, and there was no difference between groups (Figure 7B, One-way ANOVA, Tukey HSD, *p* = 0.8350). Histones co-localized with IAIPs in the nucleus, as expected (Figure 7A, IAIPs and Merged—indicated by arrows)

There was no co-localization of IAIPs with histones in the cytoplasm, as expected (data not shown), whereas cells positive for histones co-localized with cells positive for IAIPs in the nucleus of the Sham and I/R sheep (Figure 7C), and increased in number after I/R (Figure 7C; One-way ANOVA, Tukey HSD, *p* = 0.0133).

Double immunostaining with IAIPs and the endoplasmic reticulum was performed to determine the localization of IAIPs in the cytoplasmic compartment. Figure 8 contains the results of double immunostaining of the endoplasmic reticulum (anti-calnexin Ab) with IAIPs in the cerebral cortex (Figure 8A; IAIPs and Merged—indicated by arrows). Calnexin-positive cells, expressed as a percentage of the total number of DAPI-positive cells, were present in the Sham and I/R fetal sheep, and their numbers decreased after I/R (Figure 8B; One-way ANOVA, Tukey HSD, *p* = 0.0238). There were differences in the co-localization of IAIPs with the endoplasmic reticulum in the cytoplasm (Figure 8C) between groups. As expected, the calnexin staining was not detected in the nucleus (data not shown), whereas cells positive for calnexin co-localized with IAIP-positive cells in the cytoplasm of the Sham and I/R sheep (Figure 8 C), and decreased in number after I/R (Figure 8C; One-way ANOVA, Tukey HSD, *p* = 0.0280).

Taken together, these findings confirm that IAIP immunofluorescent expression is present in both the nucleus and the endoplasmic reticulum.

## 3. Discussion

The present experiments are the first to report the subcellular localization of IAIPs in the cerebral cortex and white matter of fetal sheep with and without exposure to I/R-related brain injury. In previous studies, we demonstrated the presence of 125 kDa *PaI* and 250 kDa *IaI* protein moieties by Western immunoblot in fetal sheep brains and that both were reduced in the cerebral cortex 4 h after exposure to I/R [16,27]. The current study now delineates the distribution of IAIPs in specific brain cell types, the effects of I/R on IAIPs in these cells, and the effects of I/R on changes in the cytoplasmic and nuclear localization of IAIPs after 7 days of recovery in different types of brain cells. We have validated the use of the R-22c pAb to identify IAIPs in fetal sheep plasma and brain tissue, consistent with our previous reports [16,27]. Furthermore, the current study suggests that not only are there changes in the quantity of the IAIP protein moieties by Western immunoblot analysis after I/R [27], but that the cellular localization of these proteins is also modified after I/R.

The novel findings of the current study are as follows. First, IAIPs were primarily expressed in the cytoplasmic compartment of the cerebral cortex in healthy sham control fetal sheep, whereas they were expressed mainly in the nuclear compartment after I/R. Second, IAIPs were primarily expressed in the nuclear compartment in white matter in both the Sham and I/R groups. Third, IAIPs were mainly localized to the cytoplasm in neurons in the cerebral cortex in Sham, and the percentage of cytoplasmic-positive IAIP neurons decreased after exposure to I/R. Fourth, IAIPs were localized to the nuclei of microglia in the cerebral cortex of both the Sham and I/R groups, and the percentage of total and nuclear IAIP-positive microglial cells decreased after I/R. Fifth, IAIPs were localized mainly in the nucleus of oligodendrocytes in both the Sham and I/R groups. Sixth, IAIPs were mainly localized in the nuclei of proliferating cells in both the Sham and I/R groups. Seventh, IAIPs co-localized with histones in the nucleus and the endoplasmic reticulum in the cytoplasm of the cerebral cortex.

IAIPs are gaining increasing attention because of their potential importance to inflammatory-related disorders in newborns, including sepsis, necrotizing enterocolitis, and HI-related brain injury [4,5,40,41,42]. Our previous findings suggested that ischemia resulted in reduced IAIP levels in the CNS of fetal sheep [27] and that hypoxia was associated with reduced IAIP levels in the cerebral cortex of neonatal rats [43]. In addition, we have previously found changes in the subcellular localization of IAIPs in neonatal rat brains after exposure to HI brain injury [29]. However, these studies did not define the specific brain cell types in which the cellular localization was altered by exposure to HI. In the current study, we examined whether there were specific changes in the cellular and subcellular localization of IAIPs in response to I/R-related brain injury. The subcellular localization of IAIPs was examined in three different types of cells, including neurons and microglia in the cerebral cortex, and oligodendrocytes in white matter. Remarkably, the subcellular localizations of IAIPs differed among the different cellular populations and also in response to I/R.

Similar to our findings in the human brain during development and in neonatal rats, more than 90 percent of cells in the cerebral cortex and white matter of the sheep fetus were positive for IAIPs by immunohistochemistry (Figure 2) [18,29]. These data strongly support the importance of IAIPs for normal brain function and development [16,17,18,27,29]. Interestingly, in the present study, exposure to I/R was associated with a reduced proportion of cells expressing cytoplasmic IAIPs and a reciprocal increase in the proportion of cells expressing nuclear IAIPs in the cerebral cortex. By contrast, in white matter, I/R was not associated with any change in the proportion of cells expressing IAIPs in the cytoplasm or nucleus. The reason for the divergent effect of I/R on the expression of IAIPs in the cerebral cortex and white matter is not known but emphasizes the differing regulation of IAIPs in different brain regions in the fetus.

Consistent with previous work, the number of NeuN-positive neurons decreased after I/R [35,44]. Impaired blood flow during an ischemic event reduces oxygen and glucose delivery to the brain, resulting in energy depletion, over-activation of glutamate receptors and release of excess glutamate, increase of intracellular calcium, loss of membrane potential and cell depolarization, and ultimately neuronal cell death [44]. IAIPs were localized mainly in the cytoplasm of neurons in the Sham group. The proportion of neurons expressing IAIPs in the cytoplasm decreased after I/R, whereas the proportion of nuclear IAIP-positive neurons did not change after I/R injury. This may suggest alterations in the subcellular localization of IAIPs in neurons after I/R. The cytoplasmic localization of IAIPs in neurons is consistent with our previous findings in neonatal rodents [17]. Moreover, IAIP genes were also identified in cultured neurons, suggesting the potential for local IAIP synthesis within neurons [17]. In this regard, the presence of IAIPs in the nucleus of neurons (Figure 3) could suggest the potential for neuronal synthesis of IAIPs. Likewise, we have previously reported the presence of endogenous IAIPs in the human brain over a wide range of ages, including preterm, newborn, and adult, and expressed in neurons of all age groups [18]. Given the presence of IAIPs in neurons, the decrease in IAIP expression 7 days after ischemia, combined with previously reported increases at 24 and 48 h after ischemia, and the shift in neuronal IAIP localization after I/R shown in the current study (Figure 3D), we speculate that IAIPs could be important in the neuronal response to ischemia [16,17,18,27,29]. Although we have previously shown neuroprotective effects of treatment with exogenous IAIPs after exposure to HI in neonates, we do not know if circulating IAIPs are able to cross the blood-brain barrier where they might have a direct effect on the brain [40,41,42,45,46]. Nonetheless, the neuroprotective effects of exogenous IAIPs combined with the current findings raise the intriguing possibility that endogenous IAIPs may potentially play an anti-inflammatory, neuroprotective role in endogenous neuronal recovery after ischemia. However, in spite of the ubiquity of IAIPs in normal brains and the changes after exposure to I/R, information on the endogenous function of IAIPs in the brain remains to be elucidated.

Iba1-positive microglia play an important role in enhancing the immune spectrum after insults to the brain. Interestingly, IAIPs exhibited nuclear localization in microglia in both groups. However, additional analysis is required to elucidate the basis of these findings. Consistent with previous findings, the number of Iba1-positive microglia increased after exposure to I/R (Figure 4) [38,40]. Recent studies suggest that the majority of Iba1-positive cells after brain injury are indeed microglia [47]. More than 80 percent of microglia were positive for IAIP expression in both the Sham and I/R exposed groups, but IAIP expression was higher in the Sham than in the I/R exposed fetal sheep. Although there was a smaller number of microglia with nuclear IAIP staining in the I/R group, this change appears to be driven by the lower total number of total microglia with IAIP staining.

The number of Olig2-positive oligodendrocytes decreased after I/R injury in white matter, consistent with previous reports (Figure 5) [39,48]. Almost all Olig2-positive cells expressed IAIPs exclusively in the nuclear compartment in both the Sham and I/R sheep. I/R also did not change the localization of IAIPs in oligodendrocytes. Although astrocytes are a very important form of glia in the central nervous system, and though we have previously identified IAIPs in the perinuclear area and end-feet of cultured rat astrocytes in rodents [17,49], we were not able to immunohistochemically detect subcellular localization of IAIPs in astrocytes in the fetal sheep brain.

Cellular proliferation represents an important component of the response to ischemia [30,31]. There was a modest increase in proliferating, Ki67-positive, cells in the I/R group (Figure 6), which was detected by Ki67 protein expression in the nucleus [50]. Most Ki67-positive cells were also positive for IAIP in both groups, predominantly in the nucleus (Figure 6). Therefore, cellular proliferation per se was not associated with changes in the subcellular localization of IAIPs. Although we cannot identify the cell specific proliferation because we were not able to double label the proliferating cells, the number of microglial cells increased after exposure to I/R [51], whereas the numbers of neurons and Olig2-positive oligodendrocytes decreased. Consequently, increases in nuclear staining of IAIPs after I/R injury could be related to cellular proliferation after I/R, IAIP nuclear staining having been detected both in the Ki67-positive proliferating cells and in the Iba1-positive cells, which also increased after exposure to I/R [30,31,51].

We have previously shown that IAIPs were present in the nucleus in all age groups, but that cytoplasmic IAIP expression was relatively more abundant in the adult brain [18]. We therefore questioned whether IAIPs could also be present in the perinuclear cytoplasm in the developing brain and sought to confirm the nuclear localization of IAIPs by co-staining IAIPs with histones, a well-recognized nuclear marker. The current study shows that IAIPs co-localize with histones in the nucleus in both the Sham and I/R groups and also with calnexin in the cytoplasm, suggesting that IAIPs are present in both the nucleus and cytoplasm (Figure 7 and Figure 8). In addition, IAIPs have previously been shown to neutralize the cytotoxic effects of histones during inflammatory conditions, including sepsis, to bind to recombinant histone H4, and to co-localize with histones in necrotic tissues [3].

In the current study, the percentages of neurons expressing IAIPs in the cytoplasm and of microglia expressing IAIPs in the nucleus both decreased after exposure to I/R (Figure 3D and Figure 4E). Though there is little evidence for the factors controlling cellular metabolism of IAIPs, we speculate that the reduction of IAIPs in the cellular components of neurons after I/R could represent a shift in IAIPs among subcellular localizations and potentially even their being released from the cell into the extracellular space. Recently, we reported that the subcellular localization of IAIPs shifted between the cytoplasm and nucleus after HI-related brain injury in neonatal rats, consistent with our current findings in the fetal sheep [29]. Likewise, other proteins exhibit shifts in subcellular localizations, translocation, or release from brain cells after exposure to ischemic insults [52,53,54,55]. Histone deacetylases 4 have been reported as being highly expressed in neurons, normally mainly within the cytosol [53]. However, in mice they are translocated from the cytosol into the nucleus of neurons in response to middle cerebral artery occlusion (MCAO) [53]. The high mobility group box-1 (HMGB1) damage protein is an associated-molecular pattern (DAMP) that is involved in many inflammatory-related disorders [52]. Previously, we have demonstrated HMGB1 translocation from the nucleus to cytosol in the fetal sheep brain, along with release into the extracellular space after HI in the neonatal rat brain [55]. Moreover, we have also recently shown that HMGB1 and IAIPs bind to each other in vitro and are co-localized both in the nucleus and cytoplasm after HI-related brain injury [29]. These proteins are a part of the inflammatory cascade and considering the anti-inflammatory effects of IAIPs [1,10], the current findings suggest the hypothesis that shifts in the localization of IAIPs could contribute to neuroprotective effects in HI-related inflammation in the brain.

There are several limitations to our study. Although we have shown for the first time that IAIPs show cell-specific changes in subcellular localizations after exposure to ischemia, the functional significance of these changes remains to be determined. We were not able to perform Western immunoblot analysis or ELISA on the brain samples from these studies because fresh frozen brain tissue was not available from the original studies. Finally, we cannot comment upon sex-related differences due to the limited number of animals in this fetal sheep study.

In conclusion, the present study demonstrates that IAIPs are differentially expressed at the cellular and subcellular level in the fetal sheep brain, and exposure to I/R alters IAIP subcellular localization in the cerebral cortex, neurons, and microglia, but not in proliferating cells or oligodendrocytes in white matter. We speculate that these changes in the localization of IAIPs are consistent with their playing a role in recovery from I/R in the fetal brain, which may be mediated by interactions with other proteins.

## 4. Materials and Methods

### 4.1. Study Groups, Animal Preparation, and Experimental Design

Briefly, singleton Romney/Suffolk fetal sheep were instrumented using sterile techniques at 121–124 days of gestation (full term is 145 days in this breed of sheep), as previously described [24]. The ewes were administered Streptocin (procaine penicillin (250,000 IU/mL) and dihydrostreptomycin (250 mg/mL, Stockguard Laboratories Ltd., Hamilton, New Zealand) intramuscularly for prophylaxis 30 min before the onset of surgery. Anesthesia was induced by an intravenous injection of Alfaxan (Alphaxalone, 3 mg/kg, Jurox, Rutherford, New South Wales, Australia), and general anesthesia maintained with 2–3% isoflurane in oxygen. The ewes were given constant infusions of isotonic saline to maintain fluid balance. The depth of anesthesia, maternal heart rate, and respiration were continuously monitored by trained anesthetic staff.

After a maternal midline abdominal incision, the fetus was exteriorized and both fetal brachial arteries were catheterized using polyvinyl catheters to measure mean arterial blood pressure for the original studies. An amniotic catheter was secured to the fetal shoulder. After ligation of the vertebral-occipital anastomoses, inflatable carotid occluder cuffs were placed around each carotid artery [23,35]. The uterus was sutured closed and antibiotics (80 mg Gentamicin, Pharmacia, and Upjohn, Rydalmere, New South Wales, Australia) placed in the amniotic sac. The skin laparotomy incision of the ewe was infiltrated with a local analgesic, 10 mL 0.5% bupivacaine plus adrenaline (AstraZeneca Ltd., Auckland, New Zealand). All fetal catheters were exteriorized through the flank of the ewe. The long saphenous vein of the ewe was catheterized for maternal post-operative care and euthanasia. Post-operatively, the sheep were housed together in separate cages with access to food and water ad libitum. They were kept in a temperature-controlled room with 12 h light/dark cycles. Antibiotics were given intravenously for four days to the ewe (600 mg Benzyl penicillin sodium, Novartis Ltd., Auckland, New Zealand, and 80 mg Gentamicin, Pharmacia, and Upjohn). Fetal catheters were maintained patent by continuous infusions of heparinized saline (20 U/mL at 0.15 mL/h) and the maternal catheter flushed daily.

At 128 ± 1 days of gestation (0.85 of gestation), the fetal sheep were randomly assigned to two groups: Sham occlusion-treated (Sham, *n* = 6; male = 4, female = 2), and 30 min of cerebral ischemia, induced by bilateral carotid artery occlusion, followed reperfusion and 7 days of recovery (I/R, *n* = 5; male = 2, female = 3). Ischemia was induced by reversible inflation of the carotid occluder cuffs with saline for 30 min. Control fetuses otherwise received the same treatment but the occluder cuffs were not inflated. The ewe and fetus were killed seven days after exposure of the fetus to ischemia or sham treatment with an overdose of sodium pentobarbitone (9 g I.V. to the ewe; Pentobarb 300, Chemstock International, Christchurch, N.Z.). The fetal brain was perfused with heparinized normal saline (20 IU heparin/500 mL saline) followed by 1 L of 10% neutral buffered formalin. The brains were fixed for an additional 7 days in 10% neutral buffered formalin and divided into 4 to 5 mm thick coronal sections using a sheep brain slicer matrix and embedded in paraffin (Figure 9).

### 4.2. Generation of Polyclonal Rabbit Anti-Rat IAIPs Antibody

Anti-serum against IAIPs was generated by immunizing rabbits with purified rat IAIPs with a custom polyclonal antibody production service provided by 21st Century Biochemicals, Inc (Marlboro, MA, USA). Rat IAIPs were extracted and purified from rat serum (Millipore Sigma, St. Louis, MO, USA) with an anion exchange chromatographic column (Toyopearl GigaCap Q-650M, Tosoh Bioscience, King of Prussia, PA, USA). Briefly, rat serum was diluted in 20 mM Tris-HCI + 200 mM NaCl, pH 4.2. Bound IAIPs were eluted from the column using 20 mM Tris-HCl + 750 mM NaCl, pH 6.8. Eluted proteins were buffer exchanged to phosphate-buffered saline (20 mM Phosphate + 150 mM NaCl, pH 7.4) with the Amicon Ultra Centrifugal Filter Units with regenerated cellulose membranes with molecular weight cutoffs of 30k Da (Millipore Sigma, Burlington, MA, USA). Rabbits were immunized five times with purified IAIPs (200 μg) mixed with Freund’s adjuvant. Rabbits were bled after each immunization boost for the titer analysis, and the final bleed from the rabbit was used in this study (R-22-c pAb). The reactivity of R-22-c pAb was examined by Western immunoblot analysis using purified IAIPs and serum from other species and determined that the antibody is cross-reactive with purified IAIPs and serum from human, mouse, and sheep (data not shown).

### 4.3. Validation of Purified R22-c pAb by Western Immunoblotting

Western immunoblot was used to validate the specificity of the rabbit polyclonal antibody, R-22c pAb against the rat IAIPs. Purified human IAIPs (0.016 μg), sheep serum (1:200), and sheep brain extraction (250 μg) were loaded to serve as positive and negative control on SDS-polyacrylamide gel (NuPAGE^TM^ 4–12% Bis-Tris Midi Gel, Thermo Fisher Scientific, Waltham, MA, USA) and electrophoresis was conducted for 90 min. The proteins on the gel were then transferred onto one polyvinylidene difluoride membrane (PVDF, 0.2 μg, Bio-Rad Laboratories, Hercules, CA, USA) using a semi-dry transfer technique. Thereafter, one half of the membrane was incubated with pre-immune serum (PIS, 1:3000, ProThera Biologics Inc., Providence, RI, USA) as a negative control for the R-22c pAb, whereas the other half of the membrane with the same loading proteins was incubated with R-22c pAb (1:3000, ProThera Biologics Inc.) overnight at 4 °C. After washing three times with TBS-T/0.1% Tween-20, the membrane was incubated with peroxidase-labeled goat anti-rabbit secondary antibody (1:20,000, Alpha Diagnostic, San Antonio, TX, USA) for one hour at room temperature. After incubating the membrane with enhanced chemiluminescence (ECL plus, Western Blotting Detection Reagents, Amersham Pharmacia Biotech, Inc., Piscataway, NJ, USA), the images were acquired by the ChemiDoc XRS+ System (Bio-Rad Laboratories.). The amounts of each sample loaded on both PIS- and R22-c pAb-incubated membranes were the same and they were exposed for the same amount of time.

### 4.4. Immunohistochemistry

Fixed fetal cerebral cortical tissues were sectioned at 10 µm (Leica Cryocut 1800, Leica Biosystems, Danvers, MA, USA). Sections were mounted on Superfrost Plus slides (Fisher Scientific, Co., Pittsburg, PA, USA) and stored at −80 °C until analysis. The fixed brain sections were placed in 4% PFA for 10 min at room temperature. Immunohistochemical staining was conducted as previously described [17]. Briefly, the slides were incubated in an oven at 60 °C for 1 h, and after deparaffinization and rehydration, the tissue sections were heated and pressurized for 20 min in a citrate-based antigen unmasking solution (Vector Laboratories, Burlingame, CA, USA) for antigen retrieval. Then, the slides were placed in a humidified chamber and blocked with Superblock T20 Blocking Buffer (Thermo Fisher Scientific) for 2 h after three washes with PBS at 5 min intervals. The slides were then incubated at 4 °C overnight with anti-IAIPs antibody (R-22c, 1:200, ProThera Biologics, Inc.). On the second day, after three washes with PBS at 5 min intervals, the slides were incubated at 4 °C overnight with appropriate primary antibodies, as follows: mouse anti-Fox3/neuronal nuclei (NeuN) monoclonal antibody (1:200, Abcam, Cambridge, MA, USA), goat anti-ionized calcium binding adaptor molecule 1 (Iba-1) mAb (1:1000, Abcam), goat anti-olig2 polyclonal antibody (1:500, R and D systems, Minneapolis, MN, USA), mouse anti-Ki67 mAb (1:500, Agilent, Santa Clara, CA, USA), mouse anti-histone H3 monoclonal antibody (1:500, Abcam), rabbit anti-calnexin antibody (Alexa Fluor 594 conjugated) (1:200, Abcam). Appropriate secondary antibodies were applied the next day as a cocktail at a dilution of 1:1000 after three washes, with PBS at 5 min intervals, and incubated for 1 h at room temperature. The immunoreactions were visualized with DAPI (Vectashield anti-fade mounting medium (DAPI H-1200, Vector Laboratories) after three washes with PBS at 5 min intervals. Pre-immune serum (PIS, 1:200, ProThera Biologics, Inc.) was obtained from a rabbit before immunization with rat IAIPs and used as a negative control for the specificity of the polyclonal rabbit anti-rat IAIPs antibody. All immunostaining was completed during one experimental time period and included appropriate negative controls to avoid any variability that could have arisen from different experimental conditions.

### 4.5. Image Acquisition

A Zeiss Axio Imager M2 imaging microscope system (Carl Zeiss, Inc., Jena, Germany) and ORCA-Flash4.0 LT+ Digital CMOS camera (Hamamatsu, Hamamatsu, Japan) connected with computer software Stereo Investigator 10.0 (MicroBrightField, Inc., Williston, VT, USA) was used for image acquisition for the purpose of quantification. Twelve fields within each cerebral cortical sample from each slide were first randomly selected based upon the DAPI channel. The randomly selected RGB images of the cerebral cortex were acquired with cy3 (Red), FITC (Green), and DAPI (Blue) filters, using a 40× objective from the Zeiss Axio Imager M2 imaging microscope system (Figure 2) and ORCA-Flash4.0 LT+ Digital CMOS camera (Figure 3, Figure 4, Figure 5, Figure 6, Figure 7 and Figure 8). Each wavelength was acquired separately using the same camera settings for each channel and then pseudo-merged into an RGB image for analysis in Adobe Photoshop. The optical conditions were rigorously maintained the same for all the analysis sessions.

### 4.6. Cellular Quantification

The stained slides were examined without knowledge of the experimental groups. Quantification of the number of immunoreactive cells was manually counted with a count tool in Photoshop CC (Adobe). In brief, after arranging RGB images in each color channel, the IAIP-stained cells were defined by the green channel. The other categories of stained cells and molecular elements, including the neurons, microglia, oligodendrocytes, and proliferating cells, along with the histones and endoplasmic reticulum, were defined by the red channel, and the DAPI-stained nuclei defined by the blue channel. DAPI staining was used to localize and quantify the cellular nucleus and cytoplasm. Cytoplasmic IAIP-positive cells were defined as cells with IAIP-positive staining around the nucleus in the cytoplasmic compartment with or without positive staining within the nuclear compartment (Figure 2A,C). Nuclear-positive IAIP cells were defined as cells with DAPI-positive nuclei viewed through the blue channel co-localized with the green channel that were considered nuclear IAIP-positive (Figure 2A,D). The total number of IAIP-positive cells (Figure 2A,B) was the sum of the cytoplasmic (Figure 2A,C) and nuclear (Figure 2A,D) IAIP-positive cells.

The double-stained neurons (Figure 3), microglia (Figure 4), oligodendrocytes (Figure 5) and proliferating cells (Figure 6), were first defined by the DAPI-stained nuclei and counted manually in Photoshop. Thereafter, the immunoreactivity of the neurons, microglia, oligodendrocytes, and proliferating cells were counted and the presence of cytoplasmic or nuclear IAIP immunoreactivity determined within each cell type. Similarly, the total number of cells were defined by the DAPI-stained nuclei and counted manually in Photoshop and the cells that stained for histones (Figure 7) and endoplasmic reticulum immunoreactivity (Figure 8) were also counted. Thereafter, the number of cells in which the IAIPs were co-localized with histones in the nucleus and with the endoplasmic reticulum in the cytoplasm were determined. The co-localization of IAIPs with histones in the nucleus and endoplasmic reticulum in the cytoplasm was defined by positive staining for both proteins within the same cellular location, either the nuclear or cytoplasmic subcellular compartments, as defined by the merged and overlapping fluorescent staining.

### 4.7. Statistical Analyses

Values were presented as median and scatter plots. The medians were determined for each of the samples based on the average of the cell counts from the twelve random images on each slide. Statistical analysis was performed with the Statistica (Dell Statistica, Tulsa, OK, USA) analysis program. The data in Figure 2 was analyzed using two-way analysis of variance (ANOVA) for repeated measures ANOVA, followed by the Tukey’s post-hoc test when a significant difference was detected. Comparison of two means was performed by the Two Group Student’s *t*-test with the Bonferroni correction, where appropriate in Figure 3, Figure 4, Figure 5 and Figure 6 C–E. The data in Figure 7 and Figure 8 (B and C) was analyzed using One-way ANOVA followed by the Tukey’s post-hoc test. A *p* value of *p* < 0.05 was considered as statistically significant.

## Figures and Tables

**Figure 1 ijms-22-10751-f001:**
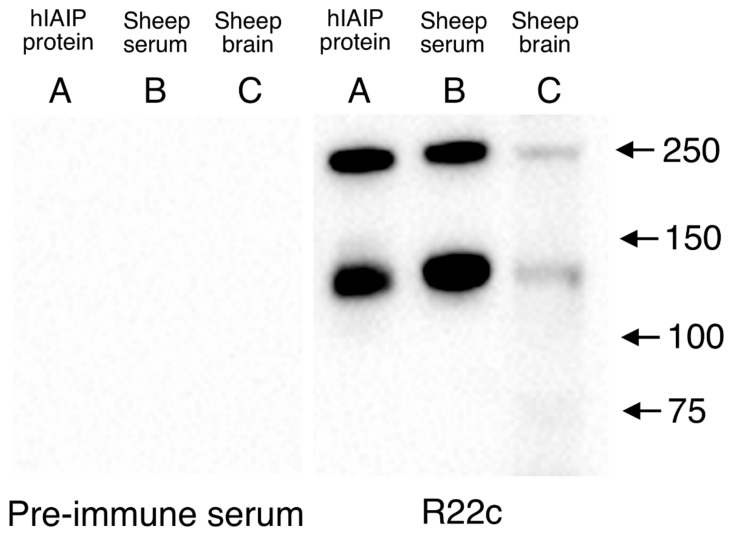
Immunohistochemical expression and quantification of IAIPs in the cerebral cortex and Western immunoblot to verify R-22c pAb specificity. Purified human IAIP protein was loaded on lane A. Sheep serum was loaded on lane B. Tissue extraction from the sheep brain was loaded on lane C. The pre-immune serum used as a negative control for the R-22C pAb did not recognize IAIP bands, whereas the R-22c pAb detected two bands, at approximately 250 and 125 kDa in the human IAIP protein, sheep serum, and sheep brain extraction (lane A, B and C).

**Figure 2 ijms-22-10751-f002:**
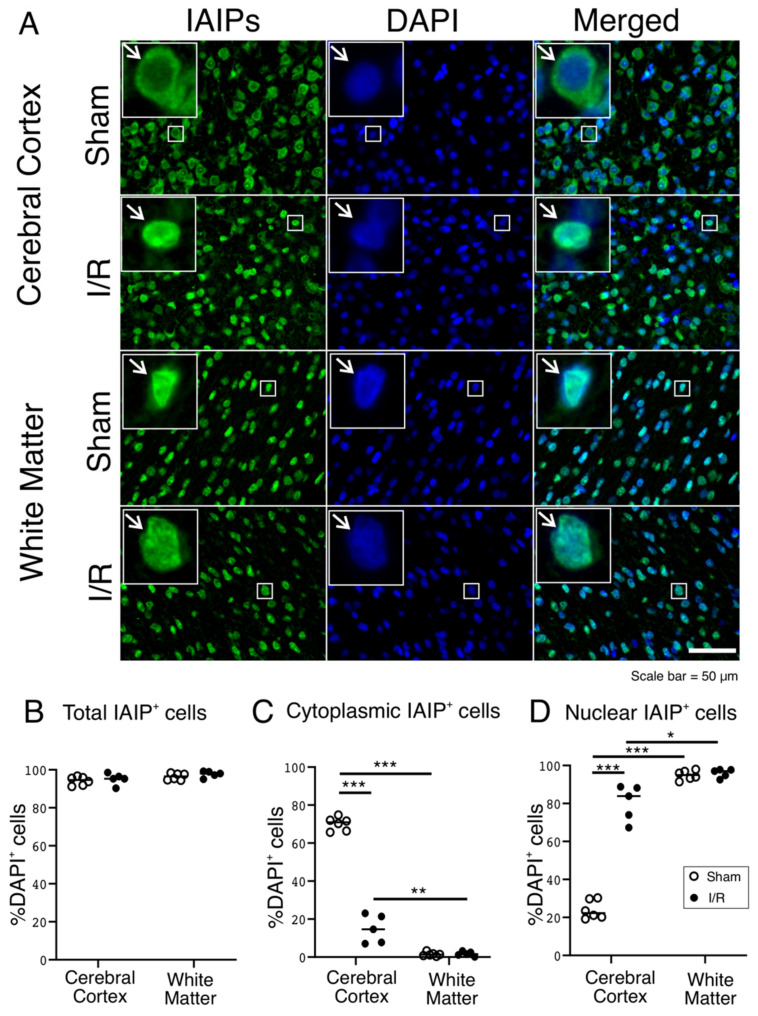
(**A**) Representative images of IAIPs (Green), DAPI-positive cells (Blue) and merged immunostaining in the cerebral cortex and white matter of fetal sheep in Sham and I/R groups. Magnification, 40×. Each inset contains high magnification images. Scale bar = 50 μm. Arrows indicate the same cell. (**B**–**D**) Quantification of IAIP-positive cells. IAIP-positive cells plotted as a percentage of DAPI-positive cells on the *y*-axis for total (**B**), cytoplasmic, (**C**) and nuclear (**D**) compartments in the cerebral cortex, and white matter plotted on the *x*-axis for the Sham (open circles) and I/R (closed circles) groups. Values median and scattergrams. ** p < 0.05, ** p < 0.01, *** p < 0.001*.

**Figure 3 ijms-22-10751-f003:**
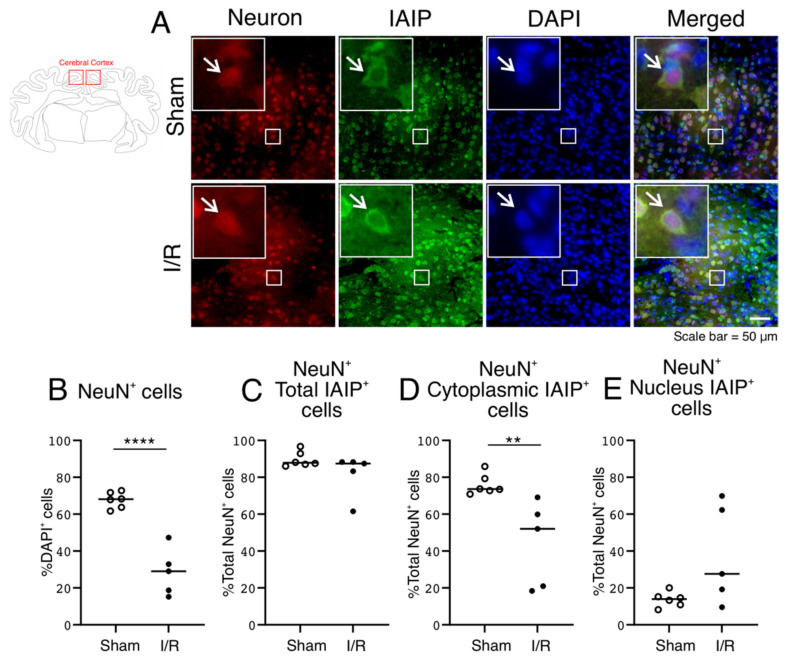
Subcellular localization of IAIPs in neurons in the cerebral cortex. Schematic diagram of a sheep brain indicates the regions from which cerebral cortical samples were taken. (**A**) Representative images of neurons (Red), IAIPs (Green), DAPI-positive cells (Blue), and merged double immunostaining in the cerebral cortex of fetal sheep. Magnification, 40×. Each inset contains the extended images. Scale bar = 50 μm. Arrows indicate the same cell. (**B**) Quantification of NeuN positively stained cells. NeuN-positive cells plotted as percentages of DAPI-positive cells on the *y*-axis for the Sham (open circles) and I/R (closed circles) groups on the *x*-axis. (**C**–**E**) Quantification of IAIP positively stained neurons. IAIP-positive NeuN-positive cells expressed as a percentage of NeuN-positive cells on the *y*-axis for total (**C**), cytoplasmic (**D**), and nuclear (**E**) compartments plotted for the Sham (open circles) and I/R (closed circles) groups on the *x*-axis. Values median and scattergrams. *** p < 0.01, **** p < 0.0001.*

**Figure 4 ijms-22-10751-f004:**
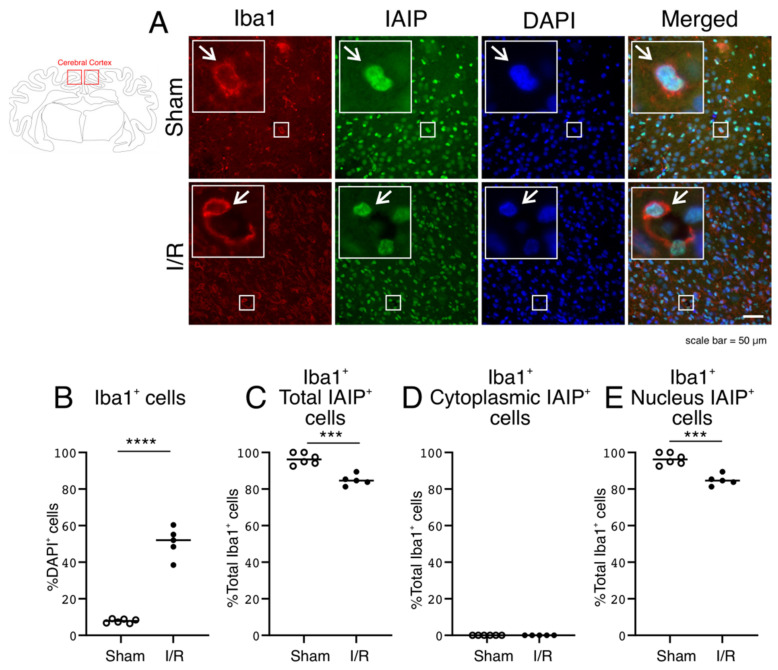
Subcellular localization of IAIPs in microglia in the cerebral cortex. Schematic diagram of sheep brain indicates regions, from which cerebral cortical samples were taken. (**A**) Representative images of microglia (Red), IAIPs (Green), DAPI-positive cells (Blue) and merged double immunostaining in the cerebral cortex of fetal sheep in Sham and I/R groups. Magnification, 40×. Each inset contains the extended images. Scale bar = 50 μm. Arrows indicate the same cell. (**B**) Quantification of Iba1 positively stained cells. Iba1 positive cells expressed as a percentage of DAPI-positive cells plotted on the *y*-axis for the Sham (open circles) and I/R (closed circles) groups on the *x*-axis. Values median and scattergrams. *** p < 0.01*. (**C**–**E**) Quantification of IAIP positive Iba1 microglia. IAIPs positive Iba1 positive microglia plotted as a percentage of Iba1 positive microglia on the *y*-axis for total (**C**), cytoplasmic (**D**), and nucleus (**E**) compartments for the Sham (open circles) and I/R (closed circles) groups on the *x*-axis. Values median and scattergrams. **** p < 0.001, **** p < 0.0001*.

**Figure 5 ijms-22-10751-f005:**
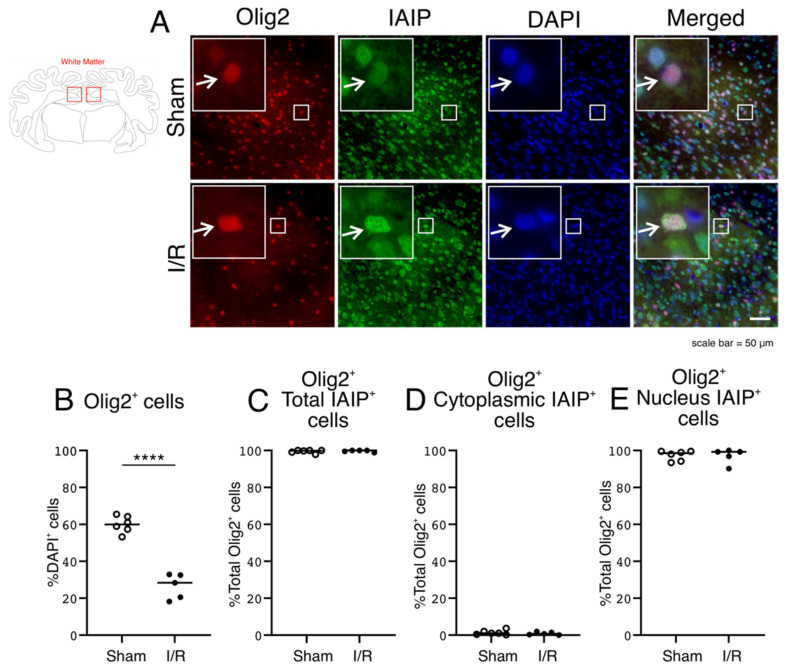
Subcellular localization of IAIPs in oligodendrocytes in white matter. Schematic diagram of sheep brain indicates regions, from which white matter samples were taken. (**A**) Representative images of oligodendrocytes (Red), IAIPs (Green), DAPI-positive cells (Blue), and merged double immunostaining in white matter of Sham and I/R groups. Magnification, 40×. Each inset contains the extended images. Scale bar = 50 μm. Arrows indicate the same cell. (**B**) Olig2 positive cells plotted as a percentage of DAPI-positive cells on the *y*-axis for the Sham (open circles) and I/R (closed circles) groups plotted on the *x*-axis. (**C**–**E**) Quantification of IAIP-positive Olig2-positive oligodendrocytes. IAIP-positive Olig2-positive expressed as a percentage of Olig2-positive cells plotted on the *y*-axis for total (**C**), cytoplasmic (**D**), and nucleus (**E**) compartments for Sham (open circles) and I/R (closed circles) groups on *x*-axis. Values median and scattergrams. ***** p < 0.0001*.

**Figure 6 ijms-22-10751-f006:**
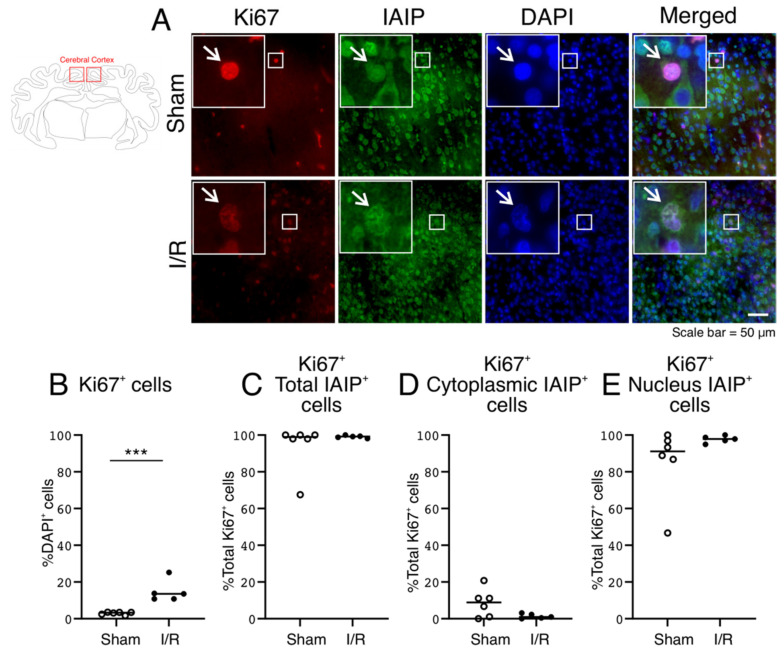
Subcellular localization of IAIPs in proliferating cells in the cerebral cortex. Schematic diagram of a sheep brain indicates the regions from which cerebral cortical samples were taken. (**A**) Representative images of proliferating cells (Red), IAIPs (Green), DAPI-positive cells (Blue) and merged double immunostaining in the cerebral cortex of fetal sheep in Sham and I/R groups. Magnification, 40×. Each inset contains the extended images. Scale bar = 50 μm. Arrows indicate the same cell. (**B**) Quantification of Ki67-positive cells. Ki67-positive cells shown as a percentage of DAPI-positive cells plotted on the *y*-axis for Sham (open circles) and I/R (closed circles) groups on the *x*-axis. (**C**–**E**) Quantification of IAIP-positive Ki67-positive proliferating cells. IAIP-positive Ki67-positive cells plotted on the *y*-axis for total (**C**), cytoplasmic (**D**), and nuclear (**E**) compartments, for Sham (open circles) and I/R (closed circles) groups on *x*-axis. Values median and scattergrams. **** p < 0.001.*

**Figure 7 ijms-22-10751-f007:**
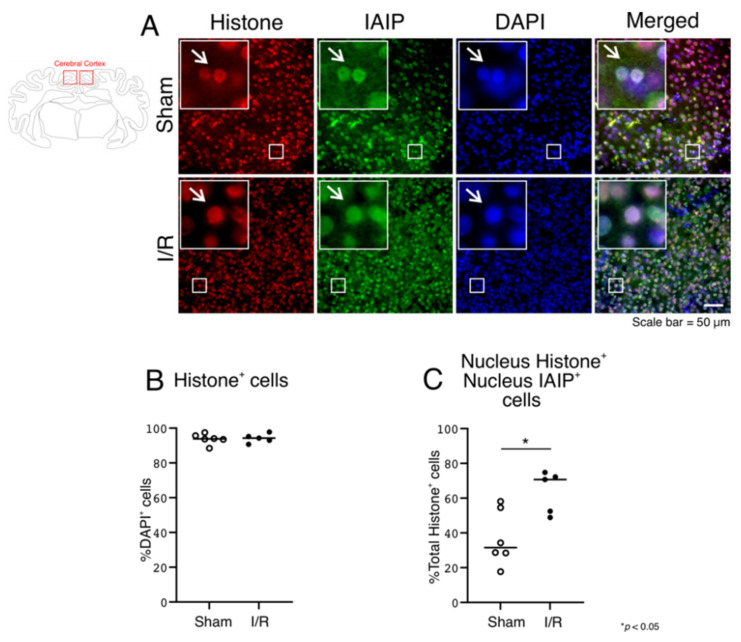
Co-localization of IAIPs with histones in the nucleus in the cerebral cortex. Schematic diagram of a sheep brain indicates the regions from which cerebral cortical samples were taken. (**A**) Representative images of histone (Red), IAIPs (Green), DAPI-positive cells (Blue), and merged double immunostaining in the cerebral cortex of fetal sheep in Sham and I/R groups. Magnification, 40×. Each inset contains the extended images. Scale bar = 50 μm. Arrows indicate the same cell. (**B**) Quantification of histone H3-positive stained cells. Histone H3-positive cells plotted as a percentage of DAPI-positive cells on the *y*-axis for Sham (open circles) and I/R (closed circles) groups on the *x*-axis. (**C**) Quantification of IAIP-positive Histone H3 nuclei. IAIP-positive histone H3-positive cells plotted on the *y*-axis for nuclear compartment for Sham (open circles) and I/R (closed circles) groups plotted on the *x*-axis. Values median and scattergrams. ** p < 0.0133*.

**Figure 8 ijms-22-10751-f008:**
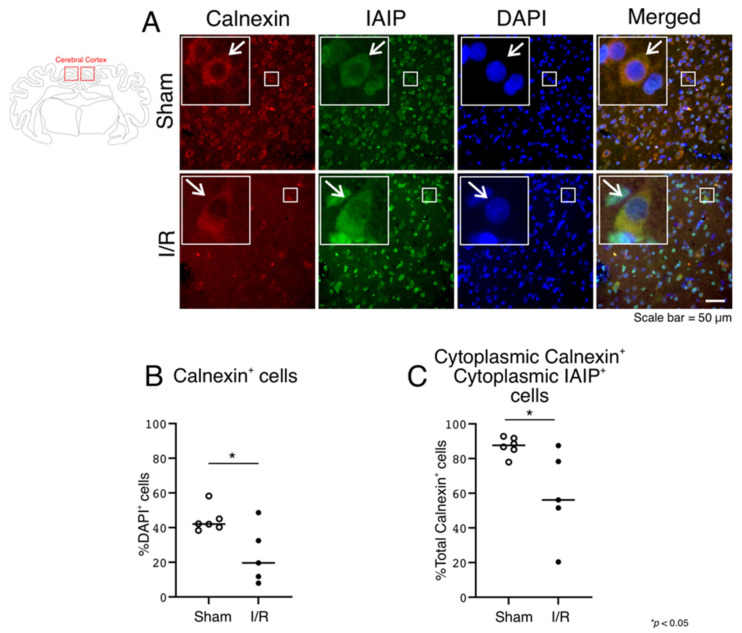
Co-localization of IAIPs with the endoplasmic reticulum in cytoplasm in the cerebral cortex. Schematic diagram of a sheep brain indicates the regions from which cerebral cortical samples were taken. (**A**) Representative images of calnexin-positive endoplasmic reticulum (Red), IAIPs (Green), DAPI-positive cells (Blue), and merged double immunostaining in the cerebral cortex of fetal sheep in Sham and I/R groups. Magnification, 40×. Each inset contains the extended images. Scale bar = 50 μm. Arrows indicate the same cell. (**B**) Quantification of calnexin-positive stained cells. IAIP-positive calnexin-positive cells plotted as a percentage of DAPI-positive cells on the y-axis for Sham (open circles) and I/R (closed circles). Values median and scattergrams. (**C**) IAIP positive and calnexin positive cells on the y-axis plotted as a percentage of total calnexin-positive cells for cytoplasmic compartments for Sham (open circles) and I/R (closed circles) groups plotted on the x-axis. Values median and scattergrams. ** p < 0.0280*.

**Figure 9 ijms-22-10751-f009:**
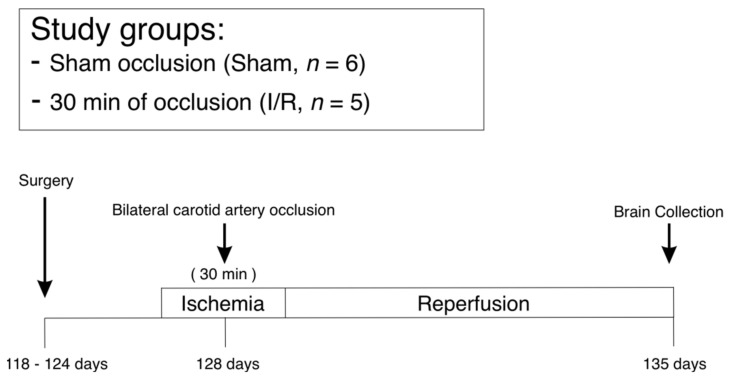
The study groups, animal preparation, and experimental design shown schematically. Fetal sheep at 128 days of gestation (85% of gestation) were randomly assigned to two groups: Sham-treated (Sham, *n* = 6), and 30 min of cerebral ischemia followed by 7 days of reperfusion (I/R, *n* = 5). Surgery was performed on the fetal sheep at 118–124 days of gestation. Occluders were placed bilaterally around the carotid arteries. At 128 days of gestation, reversible bilateral carotid artery occlusion was performed for 30 min to induce cerebral ischemia. Thereafter, the occluders were deflated and reperfusion continued for 7 days. In contrast, the occluders were not inflated in the sham group. After in vivo perfusion, the brain was collected at 135 days of gestation.

## Data Availability

Further information regarding the resources, reagents and data availability should be directed to the corresponding author and will be considered upon request.

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
