# Peer review of "Changes in Cellular Localization of Inter-Alpha Inhibitor Proteins after Cerebral Ischemia in the Near-Term Ovine Fetus"

_ijms, 2021, doi:10.3390/ijms221910751_

Round 1

Reviewer 1 Report

This manuscript describes a characterization of inter-alpha inhibitor protein (IAIP) expression in fetal sheep brain after prenatal ischemia/reperfusion injury. The authors find that there are alterations in cellular and subcellular localization of these proteins in a cell-type specific way. The manuscript is well written, and the experiments well designed. I have minor concerns about some of the statistical analysis, and the manuscript would benefit from a couple clarifications. That said, the manuscript does overall advance the field in understanding changes in expression patterns of IAIP after fetal brain injury.

- Introduction and/or discussion: There is much discussion about expression patterns of IAIPs, but it is not clear what, if anything, is known about the function of these proteins in normal brain. The paper would benefit from a little commentary about this, or at least to say that their function is not known if that is the case.

- Figure 1. The cropped blots are representative of the full blot as provided in the unpublished information.

- Results, Figures 3, 4, 5, 6 parts C, D, and E: The results in these appear to be analyzed by 2-way ANOVA with repeated measures, using total IAIP, cytoplasmic IAIP, and Nucleus IAIP as the repeated factors. This approach makes me uncomfortable because these three groups (total, cytoplasmic, nuclear) in these cases are not independent groups; any of these is defined when the values for the other two are known. It might be better from a statistical standpoint to use individual t-tests for each of these rather than the ANOVA design.

- For the data in Figures 7 and 8, I think it is technically correct to use an ANOVA design, although it does seem a bit odd to me due to the absence of histone staining in the cytoplasm and calnexin in the nucleus, respectively (basically each of these will have a group of zeroes in the analysis).

- The authors make a bigger deal than is probably warranted about the co-localization between IAIPs and histone or calnexin. This part of the manuscript is valuable as a secondary proof that staining is nuclear or cytoplasmic. However, based on the quality of the images, I don’t believe that there is sufficient resolution in the microscopy shown to really say that IAIP really co-localizes with either histone or calnexin per se. This is mainly important in the discussion, lines 446-447: I don’t believe the data are sufficient to assert that IAIPs interact with histones in the nucleus.

- Discussion, line 479: I don’t think the data warrant the conclusion that ischemia/reperfusion alters the subcellular localization of IAIPs in microglia. Although there was a smaller number of microglia with nuclear IAIP staining in the I/R group, this appears to be entirely driven by a lower total number of microglia with IAIP staining. There was no cytoplasmic staining in either the sham or I/R group of microglia, so it is hard to argue that there is a change in localization rather than just a decrease in the number of microglia expressing IAIPs at all.

Minor comments:

- The use of semicolons in places that are appropriate for commas (in the Abstract) is a little distracting.

- The paper title under Citation: (left margin, first page) has been transposed.

- Results, section 2.1, line 103. It appears that an alpha instead of the letter a is used in abbreviations IaI and PaI. Is this intentional (I don't see these abbreviations in this way elsewhere in the manuscript)?

Author Response

Reviewer #1

General comments:

This manuscript describes a characterization of inter-alpha inhibitor protein (IAIP) expression in fetal sheep brain after prenatal ischemia/reperfusion injury. The authors find that there are alterations in cellular and subcellular localization of these proteins in a cell-type specific way. The manuscript is well written, and the experiments well designed. I have minor concerns about some of the statistical analysis, and the manuscript would benefit from a couple clarifications. That said, the manuscript does overall advance the field in understanding changes in expression patterns of IAIP after fetal brain injury.

Response:  We have revised the statistical analysis as suggested. We appreciate the favorable remarks by reviewer 1.

Introduction:

-Introduction and/or discussion: There is much discussion about expression patterns of IAIPs, but it is not clear what, if anything, is known about the function of these proteins in normal brain. The paper would benefit from a little commentary about this, or at least to say that their function is not known if that is the case.

Response: The reviewer is correct. Absolutely nothing is known about the function of endogenous IAIPs in the brain. We are very interested in this area. However, this is a very difficult question to answer as the whole IAIP molecule cannot be knocked out of an animal because they are made by three chromosomes and 4 genes. There is a bikunin KO but unfortunately it is not accessible to us. We are currently attempting to perform some in vitro experiments to partially answer question. However, we have determined that IAIPs binds to High Mobility Group Box-1 in vitro and co-localizes in cerebral cortex after hypoxia-ischemia (Hatayama et al., 2021). High Mobility Group Box-1 is a damage-associated molecular patterns (DAMPs).  We have added these points to the introduction (Page 2, Line 13) and discussion (Page 13. Line 4). However, we have limited our comment as we do not have definitive knowledge of the roles of IAIPs in the brain.

- Figure 1. The cropped blots are representative of the full blot as provided in the unpublished information.

Response: This was submitted.

- Results, Figures 3, 4, 5, 6 parts C, D, and E: The results in these appear to be analyzed by 2-way ANOVA with repeated measures, using total IAIP, cytoplasmic IAIP, and Nucleus IAIP as the repeated factors. This approach makes me uncomfortable because these three groups (total, cytoplasmic, nuclear) in these cases are not independent groups; any of these is defined when the values for the other two are known. It might be better from a statistical standpoint to use individual t-tests for each of these rather than the ANOVA design.

Response: We have revised our statistical analysis as suggested and revised Figures 3, 4, 5, 6 parts C, D, and E accordingly.

- For the data in Figures 7 and 8, I think it is technically correct to use an ANOVA design, although it does seem a bit odd to me due to the absence of histone staining in the cytoplasm and calnexin in the nucleus, respectively (basically each of these will have a group of zeroes in the analysis).

Response: We have also revised the statistics for Figures 7 and 8 accordingly.

- The authors make a bigger deal than is probably warranted about the co-localization between IAIPs and histone or calnexin. This part of the manuscript is valuable as a secondary proof that staining is nuclear or cytoplasmic. However, based on the quality of the images, I don’t believe that there is sufficient resolution in the microscopy shown to really say that IAIP really co-localizes with either histone or calnexin per se. This is mainly important in the discussion, lines 446-447: I don’t believe the data are sufficient to assert that IAIPs interact with histones in the nucleus.

Response: We now provide images with greater resolution (Figure 7 and 8). However, we have also limited our discussion that IAIPs interact with histones in the nucleus (Page 13, Line 46).

- Discussion, line 479: I don’t think the data warrant the conclusion that ischemia/reperfusion alters the subcellular localization of IAIPs in microglia. Although there was a smaller number of microglia with nuclear IAIP staining in the I/R group, this appears to be entirely driven by a lower total number of microglia with IAIP staining. There was no cytoplasmic staining in either the sham or I/R group of microglia, so it is hard to argue that there is a change in localization rather than just a decrease in the number of microglia expressing IAIPs at all.

Response: We have revised the discussion to state that although there was a smaller number of microglia with nuclear IAIP staining in the I/R group, this appears to be entirely driven by a lower total number of microglia with IAIP staining as suggested (Page 13, Line 14).

Minor comments:

- The use of semicolons in places that are appropriate for commas (in the Abstract) is a little distracting.

    Response: This has been corrected.

- The paper title under Citation: (left margin, first page) has been transposed. Response: Corrected.

- Results, section 2.1, line 103. It appears that an alpha instead of the letter a is used in abbreviations IaI and PaI. Is this intentional (I don't see these abbreviations in this way elsewhere in the manuscript)?  Response: Corrected.

Reference:

Hatayama, K., Chen, R. H., Hanson, J., Teshigawara, K., Qiu, J., Santoso, A., . . . Stonestreet, B. S. (2021). High-mobility group box-1 and inter-alpha inhibitor proteins: In vitro binding and co-localization in cerebral cortex after hypoxic-ischemic injury. FASEB J, 35(3), e21399. doi:10.1096/fj.202002109RR

Reviewer 2 Report

The manuscript "Changes in cellular localization of inter-alpha inhibitor proteins after cerebral ischemia in the near-term ovine fetus" explore the subcellular translocation of IAIP in neurons and microglia in I/R. Overall the manuscript is interesting regarding the translocation of IAIP in neuronal and non-neuronal cells. 

Could author provide a high resolution images for Figure 4 and Figure 8?

IBA1 positive cells play an important role in enhancing the immune spectrum following any insult on the brain. Around 80% cells observed in I/R group are IBA1+ IAIP+ cells and almost all of them have nuclear localization. Could authors explain this more clearly in the manuscript?    

Author Response

Reviewer #2

Could author provide a high-resolution images for Figure 4 and Figure 8?

Response: We have provided high resolution images as requested.

IBA1 positive cells play an important role in enhancing the immune spectrum following any insult on the brain. Around 80% cells observed in I/R group are IBA1+ IAIP+ cells and almost all of them have nuclear localization.  Could authors explain this more clearly in the manuscript?    

Response: We have clarified this in the results and discussion (Line 13. Line 7).